# Study Protocol on Social Return on Investment (SROI) Project of the Surgical Waiting List Management System

**DOI:** 10.3390/healthcare11060825

**Published:** 2023-03-10

**Authors:** José Luis González Muñoz, Nuria García-Agua Soler, Antonio J. García Ruiz

**Affiliations:** 1Program of Biomedicine, Translational Research and New Health Technologies, School of Medicine, University of Malaga, Blvr Louis Pasteur, 29010 Malaga, Spain; 2Biomedical Research Institute of Malaga, IBIMA-Bionand, Severo Ochoa 35, 29590 Malaga, Spain; 3Department of Pharmacology and Pediatrics, School of Medicine, University of Malaga, Blvr Louis Pasteur, 29010 Malaga, Spain

**Keywords:** clinical trials, trial registration, results reporting, social return on investment (SROI), surgical waiting list, health management

## Abstract

In Andalusia, the right to maximum waiting times for healthcare clashes with the available supply, leading to an increase in demand in the form of waiting lists. To address this situation, the activity of private centers has been created for certain diagnostic tests. The Social Return on Investment (SROI) model evaluates an intervention from an economic and stakeholder perspective. However, there are no studies on the suitability of waiting lists using SROI, which is why it is intended to be studied as a decision-making tool for the clinical and healthcare management of waiting lists. This research protocol is designed to determine the quality of life gained, with the EuroQol-5D-5L questionnaire, and its social assessment, with the specific survey of the SROI method, and, thus, analyze the social return on investment and determine the suitability of the intervention (diagnostic endoscopy activity arranged in a contracted center). After the study, we will know the economic (cost in public health centers and the incremental cost of extraordinary health resources), social (quality of life with health), and environmental scenarios of the concerted activity intervention in order to adjust waiting list times.

## 1. Introduction

Waiting patients are defined as those who have been prescribed a non-urgent health intervention (mainly diagnostic or surgical) and have not yet undergone it [1], giving rise to waiting lists. This definition excludes any intervention considered urgent. An excess of demand, compared to the available supply, leads to an increase in waiting lists [2], with diagnostic and surgical procedures being the most in demand [3,4,5]. In fact, due to the COVID-19 pandemic, waiting lists have increased by 7% to 14% [6]. However, we cannot (from a management point of view) reach a zero-waiting-list commitment, as this would imply a high immediate availability of resources and that these would be idle in periods of lower demand [7,8,9]. This is why there must be a commitment to care with the shortest possible delay, within existing limitations [10,11,12,13,14]. Healthcare management in Spain is based on a decentralized model, managed by the Autonomous Communities, to optimize resources according to the population served, technology, etc. In Andalusia, the Andalusian Public Health System (APHS) is the organization that encompasses the bodies responsible for the provision and management of health services, its main provider being the Andalusian Health Service, created in 1986, which operates as an administrative agency attached to the Ministry of Health and Consumer Affairs of the Andalusian Regional Government [15,16,17]. We currently find the right to guaranteed response times for surgical interventions, first consultations, and diagnostic procedures. These establish the maximum times for being attended to, being between 90 and 180 days for certain surgical tests; 60 days for first consultations in specialized care, derived from primary care; and 30 days for a diagnostic procedure [18,19,20,21]. In recent years, the figure for concerted activity with private entities has been incorporated, to achieve health policy objectives, especially in terms of response times, constituting 4.2% of the total budget of the Andalusian Health Service (SAS) in 2020 and focusing mainly on diagnostic tests, surgical activity, bed rest, therapies, and health transport [22]. Patient referral is managed and controlled by the public hospitals through the Provincial Management Units (Order of 4 June 1998 and Resolution of 18 March 1999) of each Delegation of Equality, Health, and Social Policies. The link with the contracted centers is made through public tender, according to the criteria of the Technical and Administrative Specifications.

At present, the maximum waiting time for a diagnostic endoscopy is set at 30 days, and at the Hospital Punta de Europa, it was 86 days in the year 2020. As a strategy to achieve compliance with the maximum guarantee periods, public tenders are called for the performance of procedures close to the expiry of their deadlines. This new intervention, in the form of concerted care, requires an economic evaluation, both for the managers of the Andalusian Public Health Service, so that they can assess the suitability of the proposals presented, and for the managers of the centers that present their bids to the technical contracting committee, to present proposals that are profitable for their private centers [23]. Therefore, any intervention must be associated with an evaluation of its possible impacts, which may be economic (usefulness, effectiveness) or social (social return on investment), and which must be measured solidly so that they serve as recommendations for managers in their decision-making. Social Return on Investment (SROI) is a holistic economic evaluation method (social, economic, and environmental) used to assess impacts on different public or private policies [24], services, and interventions, which was not possible until now with the more usual forms of economic evaluation [25,26]. A cross-sectoral approach is applied, allowing all stakeholders to participate, and capturing the social value [27].

There are numerous studies in which SROI assessment has been performed, such as in psoriasis [28] or in the Management of Chronic Obstructive Pulmonary Disease [29], to cite two examples.

The aim of this type of study is to bring about a transformative change based on the modification of healthcare organizations, reducing waiting times for diagnosis and treatment in certain tests, by introducing the possibility of carrying out the tests in a contracted private hospital, where the tests will be performed. This will benefit all the agents involved, including patients, who will be placed at the center of the system.

SROI helps us to understand the preferences and needs of stakeholders, leading to improvements in decision-making that increase the value added for each stakeholder. They also explain the impact of the project by providing measures and indicators. All of this will help us to make decisions as managers, as we can assess whether we have achieved the desired impact or where we need to make an impact.

The main objectives to be studied are (i) to determine the economic cost of the interventions due to the extraordinary health concert and the ordinary activity of the Andalusian Public Health System (APHS) of the endoscopy diagnostic waiting list; (ii) to determine the increase in extraordinary resources to reduce waiting lists in terms of human and material resources; and (iii) to determine the social profitability of each endoscopy diagnostic procedure, to know, quantify, and communicate the measurement of the social return on investment in diagnostic endoscopy waiting lists in the Gibraltar West Health Area.

Other objectives of this research are:To find out the increase in extraordinary resources to reduce waiting lists in terms of human and material resources.To describe the socio-demographic data and the prevalence of surgical interventions and diagnostic tests in the extraordinary agreement and the Andalusian Public Health System (APHS).To find out how the patient’s life is affected in different areas.To determine the patient’s willingness to pay to improve their quality of life.To determine the social profitability of the proposed intervention.

This research aims to document, in the absence of previous studies, how the SROI, as an economic valuation involving all stakeholders, can be presented as a decision-making tool [27,30] for health managers and administrators on the appropriateness of healthcare agreements, in this case comparing an endoscopy intervention, between the concerted activity in private entities and the Andalusian public health system, to improve waiting lists and the impact on health-related quality of life expressed by patients.

## 2. Materials and Methods

The main objective of the study is to carry out an SROI (Social Return on Investment) project as a decision-making tool for the clinical and healthcare management of waiting lists in the Campo de Gibraltar. In addition, the researchers consider it necessary to carry out this research study, as there is no specific bibliography in this field, within the management of waiting lists.

A cross-sectional and prospective design was carried out between April 2021 and March 2022. A total of 96 patients on the waiting list for surgery and diagnostic tests at the Hospital Punta de Europa, located in the Campo de Gibraltar West health management area, and referred to the Clínica Virgen del Rosario, participated in the study. The inclusion criteria were to undergo surgery and diagnostic tests in the Hospital Punta Europa in ordinary activity and subsequently be referred for surgery and diagnostic tests, in an external health agreement, in the Clínica Virgen del Rosario, Algeciras.

Exclusion criteria were urgent surgery or urgent diagnostic tests, patients who have signed a revocation of informed consent, absence at the surgical appointment (non-cooperation), resolution of the disease causing the surgical appointment, no presentation of the disease, and refusal of the doctor or allergies.

The allocated patients were all those who were referred from the Hospital Punta Europa and who fulfilled the inclusion criteria; all of them were previously informed about the study and signed the informed consent form. The personal data of the patients were hidden and assigned a specific number to list them in the study. Patient allocation, enrolment, and participation were carried out by the principal investigator. The data are stored and kept by the principal investigator. The patients who are referred to the Virgen del Rosario Clinic are given a clinical history, with a random number, and they are given the questionnaires that they will hand in later; the researcher carries out this follow-up through the number of the clinical history.

Finally, there will be no blind patients in the study, as all patients referred and meeting the inclusion criteria will be operated on. Therefore, no unblinding procedure will be required if necessary. It is reported that no data have been collected from any other trials.

### 2.1. SROI

SROI is a framework for measuring and accounting for value more broadly, aiming to reduce inequality and improve well-being by incorporating social, economic, and environmental costs and benefits into the assessment. It also tells the story of how change is created and measures how it affects the people and organizations that experience or contribute to it.

The SROI phases are [31]:(a)Establish the scope of the assessment and the stakeholders involved.(b)Create the impact matrix.(c)Evidence of the outcomes (develop indicators) and give them an economic value.(d)Establish impact (apply correctors).(e)Calculate SROI.(f)The report, use, and communication.

The interest groups, or stakeholders, are those affected or those who can make changes. In our case, it is made up of patients (including relatives and careers), healthcare professionals (both public and private), those responsible for the public healthcare system (including management and waiting list management), and those responsible for the private healthcare center (management).

The impact of the action will be estimated using the social return on investment (SROI) methodology, with the following characteristics:(a)Action: carrying out diagnostic and therapeutic tests in a private hospital with an agreement with the reference public hospital.(b)Time horizon: 1 year.(c)Prospective analysis, to predict the future value that will be created if the action is implemented and achieves the desired results.(d)Study criteria: care, economic, and social.

The resources, or inputs, assigned will be of the human resources type, with the staff of the public or private center required to carry out the action and structural resources, such as the necessary physical spaces, and material resources, necessary for the procedures.

The outputs are related to the diagnostic activity itself, being of a healthcare, economic, and social nature, affecting the stakeholders (as they are proposed by them as reference data), and creating the necessary indicators for their measurement.

Several outcomes are envisaged, the most important of which, and the focus of this project, is the reduction of waiting lists for therapeutic or diagnostic procedures, and associated with this, a reduction in the anxiety generated by the delay in carrying out these tests, which can be associated with slowing down the necessary diagnosis and treatment. We must include family members and caregivers, who can already help to reduce or increase this anxiety.

In the contracted private hospital, we will have to assess the (theoretically negative) impact of the increased workload, initially for the healthcare staff, and in a second review for other service users. Finally, in the referral hospital, there will be a freeing up of schedules to improve their work lists.

The main outcome measure is the social return on investment, measured in the SROI ratio, which is the quotient between the total impact and the investment made. Outcomes are measured in deadweight, displacement, and attribution. For the calculation of deadweight, we can extract the data from the history of related actions, and we can extract the willingness to pay to avoid waiting (as a method for ascertaining stated patient preferences by means of contingent valuation) by means of a form prepared for this purpose. The same reasoning can be applied to workload increase and/or worklist reduction. Intangible outcomes are also included, which are quantified through the use of proxy variables.

Therefore, we will have to calculate the financial value of the investment and, on the other hand, the financial value of the social costs and benefits.

The formula used for the calculation is:$$\mathrm{ROI}\;\mathrm{ratio}=\frac{\mathrm{Present}\;\mathrm{Value}\;\left(\mathrm{Total}\;\mathrm{Present}\;\mathrm{Value}\;\mathrm{of}\;\mathrm{the}\;\mathrm{impact}\right)\;}{\mathrm{Inversión}\;\mathrm{totValue}\;\mathrm{of}\;\mathrm{the}\;\mathrm{inputs}\;\left(\mathrm{or}\;\mathrm{total}\;\mathrm{investment}\right)}$$

Any score above 1 indicates a higher social return than the investment made.

### 2.2. EuroQol-5D-5L

EuroQol-5D-5L is a validated and generic form, not referring to any specific disease, which presents results on health-related quality of life and is widely used in cost–utility analyses in health economic evaluation. It, therefore, provides information on the health of the population and, most importantly, the individual’s own assessment of their personal or self-perceived health status [30]; thus, we have a health unit, Quality Adjusted Life Year (QALY).

This form is divided into two parts, the EQ-5D descriptive system itself and the Visual Analogue Scale (VAS). The EQ-5D descriptive system comprises 5 dimensions: mobility, self-care, usual activities, pain/discomfort, and anxiety/depression, with five possible responses according to the level of severity: no problems (1), mild problems (2), moderate problems (3), severe problems (4), and extreme problems/impossibility (5). In the VAS, the individual scores his or her health between two extremes, 0 and 100, the worst and the best imaginable state of health.

Its importance in this study and protocol is significant, as it allows us to have a suitable reference value to make intergroup comparisons (for example, between patients or groups studied) and to objectify the evolution of the health of patients so that measures of effectiveness can be obtained in evaluations related to health economics and health technology assessment.

### 2.3. Confidentiality

The confidentiality and safekeeping of the study material will be guaranteed by the principal investigator in charge of data collection and processing. The name or identity of the subjects will not appear in any report, result, or publication related to the study.

The patient information sheet specifies the rights of access, modification, opposition, and cancellation of data and data portability. It also informs about the responsibility of the researcher, the conservation of the information, and the possibility of contacting the Data Protection Agency (Agencia de Protección de Datos).

### 2.4. Statistical Methods

A general descriptive analysis of the variables included in the study will be carried out. The absolute and relative frequency distributions of the qualitative variables will be presented, as well as the measures of central tendency and dispersion (mean and standard deviation) of the quantitative variables. The 95% confidence intervals for the main quantitative variables will be presented.

The SROI ratio will be calculated as the quotient of the total value of the impact between the values of the total investment of the procedures under study. The variables susceptible to variation that will be included in the sensitivity analysis will be the willingness to pay and the amortization of the investments.

The researcher shall make the complete protocol, the full set of data, and statistical procedures publicly available to any person who requests them.

### 2.5. Oversight and Monitoring

Composition of the coordinating center and trial steering committee: The protocol is coordinated by the principal investigator under his/her responsibility and will be coordinated and directed by the director of the doctoral program, providing his support and supervision. The meetings will be held once a month as a priority.

Adverse event reporting and harms: There are no adverse events or harms in the conduct of the study.

Plans for communicating important protocol amendments to relevant parties (e.g., trial participants, ethical committees): If any important changes or modifications to the protocol need to be communicated, the researcher will contact the participants. In the case of any modification of the study with respect to the ethics committee of Cadiz, it will be submitted via the digital platform provided for this purpose.

## 3. Results and Discussion

### 3.1. Primary Outcome

The completion of the quality-of-life questionnaire (EuroQol-5D-5L) is to ascertain the initial state of health of the patient referred from Hospital Punta Europa. The completion of the specific survey of the SROI method is to find out specific aspects of the patient’s life related to their habits. This survey is based on areas such as self-care, emotional, social relations, family, sexuality, leisure, community, education, and work.

#### 3.1.1. Secondary Outcome

The EuroQol-5D-5L and SROI method questionnaires will be recorded.

#### 3.1.2. Independent Variables

The independent variables will be diagnoses (Table 1), therapeutic procedures (Table 2), place of birth (Table 3), gender (Table 4), costs of procedures, age, etc.

During the study period, different tasks have been carried out over different periods of time. At the beginning of the study, the explanatory document of the study was prepared for the patients together with the informed consent forms, in addition to the specific survey of the SROI methodology and the EuroQol survey (Table 5).

Once all the documentation was completed and the authorization of the research ethics committee was obtained, patients were assigned and documents were handed over to the study patients between November 2021 and March 2022. The first primary results on pathologies and procedures were obtained between April and May of the same year.

In the period under study, 96 therapeutic procedures were performed, the most performed being endoscopic polypectomy of the large bowel (n = 44, 45.8%), followed by colonoscopy without biopsy (n = 29, 30.2%). The three main post-testing diagnoses were no dysplasia (n = 23, 24%), mild epithelial dysplasia (n = 21, 22%), and hyperplastic polyps (n = 17, 18%).

Among the patients, 67.7% (65) came from the same locality as the center where the tests were carried out, with both sexes being represented in equal proportions. Tarifa is 23 km away, the furthest point from the hospital, which accounted for 10.4% of the patients (10), so distance was not an obstacle for patients to choose the option of having the test performed at an approved center.

## 4. Conclusions

In relation to the main objective of this study, we designed an SROI project as a decision-making tool for the clinical and health management of waiting lists in Campo de Gibraltar. This study will allow us to analyze in depth how patients in the Andalusian Public Health System (APHS) are affected by delays in their diagnostic or surgical procedures.

Based on the proposed action in the Campo de Gibraltar West health management area, our hypothesis is that the adoption of a decision-making tool for the clinical and health management of waiting lists is a good way to minimize the time patients have to wait for surgery, which has an impact on the quality of life in health reported by these patients.

It also allows us to obtain primary information on socio-demographic characteristics, the type of procedures to be performed, the patient’s willingness to pay to solve their illness, and to know the available or future healthcare resources for solving the problem of waiting lists.

This protocol should help hospital managers to increase the efficiency of their interventions. Therefore, we believe that in order to improve the efficiency of the national health system, without losing quality and safety, it would be advisable to consider the SROI method.

In regard to limitations, we found a reluctance of some patients to answer the surveys, perhaps motivated by mistrust of the research. To overcome this, an attempt was made to organize a single information meeting with the patients.

Another important restriction comes from the subjectivity of the research itself, as the stakeholders and patients belong to a limited health area, so the results are valid for documenting this type of intervention, but not for generalizing in the context of wider health systems. However, it is also a strength in that it can be perfectly adapted to the microenvironment in which it works.

Another limitation comes from the implementation of the SROI system, as it would require an investment, since current resources for research and the implementation of this type of intervention in the Andalusian Public Health System are scarce.

As future lines of work, we propose the evaluation of other diagnostic and therapeutic groups, preferably from different health systems, in order to extrapolate data and make it more useful for the management of waiting lists, with the possibility of concerted activity according to its efficiency.

## Figures and Tables

**Table 1 healthcare-11-00825-t001:** Includes the diagnoses of the patients surveyed.

Diagnosis	Frequency	Percentage (%)	Accumulated Percentage (%)
No dysplasia	23	24	24
Mild epithelial dysplasia	21	22	46
Hyperplastic polyps	17	18	64
Hemorrhoids	7	7	71
Non-diagnostic	7	7	78
Gastritis	6	6	84
Dolichocolon	6	6	91
Colitis	5	5	96
Other diagnoses	4	4	100

**Table 2 healthcare-11-00825-t002:** Includes the therapeutic procedures of the patients studied.

	Frequency	Percentage (%)	Valid Percentage (%)	Accumulated Percentage (%)
Esophagogastroduodenoscopy without biopsy	6	6.3	6.3	6.3
Esophagogastroduodenoscopy with biopsy	6	6.3	6.3	12.5
Large intestine endoscopy polypectomy	44	45.8	45.8	58.3
Colonoscopy with biopsy	4	4.2	4.2	62.5
Colonoscopy without biopsy	29	30.2	30.2	92.7
Esophagogastroduodenosco py without biopsy and polypectomy	1	1	1	93.8
Esophagogastroduodenosco py with biopsy and colonoscopy with biopsy	1	1	1	94.8
Esophagogastroduodenosco py with biopsy and polypectomy	2	2.1	2.1	96.9
Esophagogastroduodenosco py without biopsy and colonoscopy with biopsy	2	2.1	2.1	99
Esophagogastroduodenosco py with biopsy and colonoscopy without biopsy	1	1	1	100
Total	96	100	100	

**Table 3 healthcare-11-00825-t003:** Includes the place of residence of the patients studied.

		Frequency	Percentage (%)	Valid Percentage (%)	Accumulated Percentage (%)
Valid	Algeciras	65	67.7	69.9	69.9
	Tarifa	10	10.4	10.8	80.6
	Los Barrios	17	17.7	18.3	98.9
	San Roque	1	1	1.1	100
	Total	93	96.9	100	
Not Valid	Lost	3	3.1		
Total		96	100		

**Table 4 healthcare-11-00825-t004:** Specify the number of patients according to gender.

		Frequency	Percentage (%)	Valid Percentage (%)	Accumulated Percentage (%)
Valid	Male	47	49	50	50
	Female	47	49	50	100
	Total	94	97.9	100	
Not valid	Lost	2	2.1		
Total		96	100		

**Table 5 healthcare-11-00825-t005:** Participant timeline.

Timepoint	September 2021	October 2021	November 2021	December 2021	January 2022	February 2022	March 2022	April 2022
Enrolment SROI survey	X							
Informed consent	X							
Euroqol survey	X							
Allocation interventions		X						
Patients referred to Hospital Punta Europa			X	X	X	X	X	
Assessments Socio-demographic variables			X	X	X	X	X	X
Variables SROI methodology			X	X	X	X	X	X
Primary outcomes on pathologies and procedures							X	X

## Data Availability

All analyzed and raw data (masked to protect the information of volunteers) will be available upon reasonable request to the corresponding authors through email after the publication of the articles containing such information. A signed data access agreement will be requested to share the data. The study protocol is also available online and included as an annex to this article.

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
