# Peer review of "Study Protocol on Social Return on Investment (SROI) Project of the Surgical Waiting List Management System"

_healthcare, 2023, doi:10.3390/healthcare11060825_

Round 1
Reviewer 1 Report
Dear Authors,
criteria that guide in decision-making process include whether the manuscript represent a sufficiently profound step forward in understanding to be of broad interest to researchers outside the field, the importance of achieving balance among the topics we cover, the project's quality and completeness of the manuscript, the potential fit with planned article collections and/or the overall scope of the Journal, and interest to the readers and the general public. The potential to advance research, policy, practice, and education on medical and health issues is our foremost consideration.
I recommend that you take into consideration some aspects to improve your protocol, which are the following:
1) In the methodology you should specify what the specific steps are in the studies of an SROI 2) On the other hand, you should specify which agents will be part of the improvement proposals. 3) In addition, you should point out some of the improvements that will be proposed to achieve the best approach to the management of the waiting lists 4) You are taking euroqol as a work tool on the quality of life, but it does not specify where the source of your specific SROI survey is from. 5) Finally, in the results you should tell us what operation you will perform to get the index or value on the SROI, that is, once evaluated with which formula you will represent the result. If you modify these aspects, we will have no difficulty publishing your research protocol. Best regards, ReviewerAuthor Response
Dear reviewer, thank you very much for your comments. We have modified the manuscript following your suggestions. The changes to the original are in red for easy identification.
R: In the methodology you should specify what the specific steps are in the studies of an SROI.
A: A: We have included it following your indications. We hope it will be of interest to you
R: On the other hand, you should specify which agents will be part of the improvement proposals.
A: The interest groups, or stakeholders, are those affected or those who can make changes. In our case, it is made up of patients (including relatives and carers), healthcare professionals (both public and private), public healthcare system managers (including management and waiting list management), and private healthcare center managers (management). We have detailed this in the paper in this new revision.
R: In addition, you should point out some of the improvements that will be proposed to achieve the best approach to the management of the waiting lists.
A: This manuscript is a working protocol, so there is no publishable data yet. However, in the preparatory sessions, each stakeholder has proposed various measures to reduce surgical waiting lists, such as extending operating hours for these tests (proposed by those affected), carrying out extra activity outside normal working hours (by professionals), or greater referral to private centers (commented by the management of the private hospital). When all the results are available, we intend to publish them for the public's knowledge and as the last step in an SROI study (reporting the results).
R: You are taking euroqol as a work tool on the quality of life, but it does not specify where the source of your specific SROI survey is from.
A: The stakeholder working group shares and agrees on a series of proposals, related to the topic of work, its implications, and solutions. Once these have been established, they are evaluated economically. With them, an impact map must be created, which includes the resources necessary for the activities being analyzed (inputs), those necessary to carry them out (outputs) that will be translated into final results (outcomes) for the stakeholders and establish indicators to measure them and start with the SROI. It is more detailed in the new correction of the manuscript.
R: Finally, in the results you should tell us what operation you will perform to get the index or value on the SROI, that is, once evaluated with which formula you will represent the result.
A: The SROI ratio will be calculated as the quotient of the total value of the impact between the value of the total investment of the procedures under study. The variables susceptible to variation that will be included in the sensitivity analysis will be the willingness to pay and the amortisation of the investments. It is more detailed in the new correction of the manuscript
Reviewer 2 Report
The paper "Study Protocol on Social Return on Investment (SROI) Project of the Surgical Waiting List Management System" is interesting. However, there are some points that can be improved.
1. In the summary, I consider that it is too long. If possible reduce it and add the implications of the main results. Knowing these results helps what?
2. In the introduction, the authors must specify the contribution of the paper, what gap in the literature is filled by this paper?
3. The results and discussion section is very poor. The authors should make an effort to better present this section.
4. The conclusions must be improved. It is important that the authors highlight the main results, the contribution of the paper and the implications. In addition to adding the limitations of the study and future lines of research.
Author Response
Dear reviewer, thank you very much for your comments. We have modified the manuscript following your suggestions. The changes to the original are in red for easy identification.
R: In the summary, I consider that it is too long. If possible reduce it and add the implications of the main results. Knowing these results helps what?
A: It was indeed too long according to the journal's indications. We have revised it according to your indications. We hope it will be of interest to you.
R: In the introduction, the authors must specify the contribution of the paper, what gap in the literature is filled by this paper?
A: Thank you for your comment, we will rewrite it to add your suggestion.
R: The results and discussion section is very poor. The authors should make an effort to better present this section.
A: This manuscript is a study protocol, so there is no publishable data yet, except the initial socio-demographic study. When all the results are available we intend to publish them for public knowledge and as the last step of an SROI study (Reporting the results). We add information in the Conclusions section to show how useful an SROI study can be in the context of an economic valuation and how it can help the manager's decisions.
R: The conclusions must be improved. It is important that the authors highlight the main results, the contribution of the paper and the implications. In addition to adding the limitations of the study and future lines of research.
A: The proposal presents a Study Protocol on the Social Return on Investment (SROI) Project of the Surgical Waiting List Management System, a decision-making tool for clinical and health management of waiting lists. However, we have modified the conclusions to make the importance of this research clearer. The same is done for the limitations and future lines of research.
Reviewer 3 Report
The proposal present a Study Protocol on Social Return on Investment (SROI) Project of the Surgical Waiting List Management System, a decision-making tool for clinical and health management of waiting lists. The proposal presents an interesting topic; however, the following aspects were identified:
1. In the introduction, it is suggested to indicate the main scientific contribution, as well as the impact and benefits in the health sector in order to highlight the novelty of the proposal.
2. Likewise, it is suggested to include in the introduction a section where the works related to this proposal are indicated, indicating primarily the main differences.
3. Is the questionnaire applied to patients in a physical form or through a digital platform? Is it only addressed to the patient? Has the patient's family member's satisfaction with the patient's waiting and care been considered? Which dependent variables were considered? What are the problems for patients if there is a time that exceeds the maximum time allowed? What are the benefits of a short or very short waiting time?
4. It is suggested to include an analysis and discussion section where the authors describe the main findings identified, trends, suggestions and indicate their analysis of the work done and discuss the results.
5. In addition, it is suggested to indicate the future work to be done once the results of this study have been obtained.
6. Verify the format and style of references requested by the journal.
Author Response
Dear reviewer, thank you very much for your comments. We have modified the manuscript following your suggestions. The changes to the original are in red for easy identification.
R: In the introduction, it is suggested to indicate the main scientific contribution, as well as the impact and benefits in the health sector in order to highlight the novelty of the proposal.
A: We have included it following your indications. We hope it will be of interest to you.
R: Likewise, it is suggested to include in the introduction a section where the works related to this proposal are indicated, indicating primarily the main differences.
A: There is no literature on the use of SROI to measure the efficiency of surgical waiting lists. This was the main reason for studying this type of evaluation. We have included other health-related uses.
R: Is the questionnaire applied to patients in a physical form or through a digital platform? Is it only addressed to the patient? Has the patient's family member's satisfaction with the patient's waiting and care been considered? Which dependent variables were considered? What are the problems for patients if there is a time that exceeds the maximum time allowed? What are the benefits of a short or very short waiting time?
A: Patients are asked to fill in, on paper, the EuroQol-5D-5L form, which measures the quality of life gained. In this way we observe an initial assessment and whether or not the intervention helps the patient in the improvement of their disease, measured as quality time lived. The opinion of relatives and caregivers has been taken into account for the SROI economic analysis, as agents involved in the intervention. From the patient's point of view, a shorter waiting time means that their illness is resolved sooner, and therefore reduces anxiety, time off work, possible aggravations, etc. The importance of the SROI analysis is that it includes the non-medical part, such as patients and relatives, and they can contribute their opinion, measure it, and transfer it to an indicator that indicates the suitability of these suggestions.
R: It is suggested to include an analysis and discussion section where the authors describe the main findings identified, trends, suggestions and indicate their analysis of the work done and discuss the results.
A: This manuscript is a research protocol, and final data are not yet available. For this reason we do not present conventional results. We have added those currently available. When all the results are available we intend to publish them for the knowledge of the scientific society and as the last step of a SROI study (Reporting of results).
R: In addition, it is suggested to indicate the future work to be done once the results of this study have been obtained.
A: Following your comment we add future lines of research.
R: Verify the format and style of references requested by the journal.
A: Thank you, it did not follow any accepted format. We have changed it to APA format and corrected it.
Reviewer 4 Report
The idea of using the SROI methodology on waiting lists is interesting. However, the paper is confusing in some sections and has gaps that should be answered.
The Introduction should provide more information about SROI, its different stages, and its potential in health care, such as waiting lists.
The Materials and Methods section must be reformulated. It is currently a list of tasks to check. It does not provide the necessary link between the various phases of SROI, the variables used, the measurement instruments used and the stakeholders involved. The SROI phases can vary between 6 (1, 2) and 9 (3) steps, depending on the literature used. Understanding how the sections presented (2.1 to 2.10) fit into the SROI phases is very difficult in the Materials and Methods. In the Results and Discussion, two instruments are mentioned (EuroQol-5D-5L and the specific survey of the SROI method to find out specific aspects of the patient's life related to their habits) which in the Materials and Methods section are not even mentioned and no information is given about them either. Although EuroQol-5D-5L is a well-known tool, why was it applied? What dictated its choice? What is the specific survey of the SROI method to discover particular aspects of the patient's life related to their habits?
The Results and Discussion section is incomplete. It only presents the distribution of the independent variables used (?) and the participant's timeline. No results or discussion related to the SROI perspective – impact and investment – are presented.
In the Conclusion section, it is challenging to see the connection between the objectives proposed in the Introduction and what is said to have been done (“In this study, we conducted a questionnaire on relevant topics such as sociodemographic characteristics, type of procedures to be performed, willingness to help monetarily to improve the economic situation of the Hospital and the use of healthcare resources”). Moreover, as mentioned, there is a lack of information in the Materials and Methods and the Results and Discussion sections.
Some references should be checked. For instance, last reference (29) is incomplete. I assume you are referring to “Merino, M., Jiménez, M., Manito, N., Casariego, E., Ivanova, Y., González‐Domínguez, A., ... & Blanch, C. (2020). The social return on investment of a new approach to heart failure in the Spanish National Health System. ESC Heart Failure, 7(1), 131-138.”. This article is an excellent reference to support the reformulation of your work.
(1) A guide to Social Return on Investment (Cabinet Office, Office of the Third Sector, UK, 2009)
(2) A guide to Social Return on Investment (The SROI Network Acounting for Value, 2012)
(3) Social Return On Investment: A practical guide for the development cooperation sector (Context, international cooperation, Netherlands, 2010)
Author Response
Dear reviewer, thank you very much for your comments. We have modified the manuscript following your suggestions. The changes to the original are in red for easy identification.
R: The Introduction should provide more information on SROI, its different stages and its potential in health care, such as waiting lists.
A: We have included it following your indications. We hope it will be of interest to you
R: The Materials and Methods section must be reformulated. It is currently a list of tasks to check. It does not provide the necessary link between the various phases of SROI, the variables used, the measurement instruments used and the stakeholders involved. The SROI phases can vary between 6 (1, 2) and 9 (3) steps, depending on the literature used. Understanding how the sections presented (2.1 to 2.10) fit into the SROI phases is very difficult in the Materials and Methods.
A: We have included a new section in Material and Methods to explain how we made the choice of the SROI assessment and the steps followed.
R: In the Results and Discussion, two instruments are mentioned (EuroQol-5D-5L and the specific survey of the SROI method to find out specific aspects of the patient's life related to their habits) which in the Materials and Methods section are not even mentioned and no information is given about them either.
A; The reason is that this paper is a protocol, and the final results, which you are missing, are not yet available (our idea is to publish them for the knowledge of the scientific community). We have added the available ones, which are the sociodemographic and activity variables.
R: Although EuroQol-5D-5L is a well-known tool, why was it applied? What dictated its choice? What is the specific survey of the SROI method to discover particular aspects of the patient's life related to their habits?
A: We have included a new section in Material and Methods to explain the EuroQol-5D-5L election.
R: The Results and Discussion section is incomplete. It only presents the distribution of the independent variables used (?) and the participant's timeline. No results or discussion related to the SROI perspective – impact and investment – are presented.
A: As I mentioned earlier, this is the presentation of the protocol and we do not have this information at the time of writing the paper. We have expanded as much as possible with the information we have, especially to make the usefulness of the SROI assessment understood and how it can help managers in their decision-making.
R: In the Conclusion section, it is challenging to see the connection between the objectives proposed in the Introduction and what is said to have been done (“In this study, we conducted a questionnaire on relevant topics such as sociodemographic characteristics, type of procedures to be performed, willingness to help monetarily to improve the economic situation of the Hospital and the use of healthcare resources”). Moreover, as mentioned, there is a lack of information in the Materials and Methods and the Results and Discussion sections.
A: We have modified this part to make it more understandable. Thank you for your suggestion. R: Some references should be checked. For instance, last reference (29) is incomplete. I assume you are referring to “Merino, M., Jiménez, M., Manito, N., Casariego, E., Ivanova, Y., González‐Domínguez, A., ... & Blanch, C. (2020). The social return on investment of a new approach to heart failure in the Spanish National Health System. ESC Heart Failure, 7(1), 131-138.”. This article is an excellent reference to support the reformulation of your work.
A: Thank you for your input, we have reviewed all the references and included the suggested ones, which will give more scientific significance to the paper.
Round 2
Reviewer 2 Report
The new version of the paper "Study Protocol on Social Return on Investment (SROI) Project of the Surgical Waiting List Management System" has improved enough and I consider that it could be publishable.
Reviewer 3 Report
The authors correctly followed the indications of the previous review.